# Design of Compact Mid-Infrared Cooled Echelle Spectrometer Based on Toroidal Uniform-Line-Spaced (TULS) Grating

**DOI:** 10.3390/s22197291

**Published:** 2022-09-26

**Authors:** Qingyu Wang, Honghai Shen, Weiqi Liu, Jingzhong Zhang, Lingtong Meng

**Affiliations:** 1Key Laboratory of Airborne Optical Imaging and Measurement, Changchun Institute of Optics, Fine Mechanics and Physics, Chinese Academy of Sciences, Changchun 130033, China; 2Changchun Institute of Optics, Fine Mechanics and Physics, Chinese Academy of Sciences, Changchun 130033, China; 3University of Chinese Academy of Sciences, Beijing 100039, China; 4Forest Protection Research Institute of Heilongjiang Province, Harbin 150040, China

**Keywords:** spectroscopy, echelle spectrometer, silicon immersion grating, infrared spectroscopy

## Abstract

A traditional flat-panel spectrometer does not allow high-resolution observation and miniaturization simultaneously. In this study, a compact, high-resolution cross-dispersion spectrometer was designed based on the theoretical basis of echelle grating for recording an infrared spectrum. To meet the high-resolution observation and miniaturization design requirements, a reflective immersion grating was used as the primary spectroscopic device. To compress the beam aperture of the imaging system, the order-separation device of the spectrometer adopted toroidal uniform line grating, which had both imaging and dispersion functions in the spectrometer. The aberration balance condition of the toroidal uniform line grating was analyzed based on the optical path difference function of the concave grating, and dispersion characteristics of the immersed grating and thermal design of the infrared lens were discussed based on the echelle grating. An immersion echelle spectrometer optical system consisting of a culmination system, an immersed echelle grating, and a converged system was used. The spectrometer was based on the asymmetrical Czerny-Turner and Littrow mount designs, and it was equipped with a 320 × 256 pixel detector array. The designed wavelength range was 3.7–4.8 μm, the F-number was 4, and the central wavelength resolution was approximately 30,000. An infrared cooling detector was used. The design results showed that, in the operating band range, the root implied that the square diameter of the spectrometer spot diagram was less than 30 μm, the energy was concentrated in a pixel size range, and the spectrometer system design met the requirements.

## 1. Introduction

Spectroscopy is one of the most common gas detection methods. By measuring the exemplary characteristic spectra of molecules, atoms, or ions in the atmosphere, characteristic atmospheric parameters, such as types and concentrations of atmospheric components, can be inverted, thereby providing important primary scientific data for climate change prediction and atmospheric science research. Detecting small matter is a frontier subject area of national space high-tech, and it has always been a focus of developed countries worldwide. However, current research on detection methods in this field is mainly based on ground-based active detection (radar). The method has some limitations, including the difficulty to perform extended surveillance and the likelihood to lose the target [1]. Infrared spectroscopy has wide zone characteristics, high efficiency, all-day use, etc., and this is an innovative technological way to solve the problem of remote sensing of materials. Spectrometers are an essential part of analysis and testing instruments and play an important role in national defense construction scientific research, industrial production, and national economy development. With the improvement of grating processing and manufacturing technology and demand, the development of spectrometers with high dispersions, high resolutions, small volumes, and high intelligence is important in the spectrometer industry [2,3,4]. Ordinary blazed grating meets the requirements of luminous flux. Ordinary blazed gratings can only use lower orders (first- or second-order) to avoid order overlap in practical applications. Therefore, only large-area, thin-line blazed gratings can be used to obtain high-resolution spectra. This makes the size of the instrument tremendous [5]. However, the use of echelle grating has played an essential role in improving the characteristics of spectrometers.

In 1949, Harrison first proposed the concept of echelle grating and discussed the feasibility and superiority of using echelle grating as the main dispersive element in spectrometry, which laid the foundation for the development of the echelle grating theory [6]. It also provided a new orientation for high-resolution spectrometers. The echelle grating spectrometer is a high-resolution spectrometer that employs echelle grating with a low-dispersion transverse beam-splitting element to form a two-dimensional overlapping spectrum on the image plane. Through this spectral form, the echelle grating spectrometer effectively overcomes the heavyset weak point of traditional high-resolution spectrometers and simultaneously does not compel multiple scanning exposures and can achieve a transient, direct-reading measurement of the spectrum. Compared with classic grating spectrometers, echelle grating spectrometers have the characteristics of high resolution, low detection limit, wide wavelength band, stationary parts, compact structure, and direct reading of the full spectrum. They created a pathway for the development of the next generation of spectrometers [7,8,9,10,11,12]. Echelle spectrometers have been employed in the detection of trace substances in the atmosphere [13,14], the observation of stars [15], inductively coupled plasma spectroscopy (ICP) systems [16,17,18], laser-induced plasma spectroscopy systems (LIPSs) [19,20], etc.

Classical gratings draw on increasing the diffraction order and the total number of grating lines to achieve high resolution, but a high diffraction order leads to an increase in the grating constant, resulting in a serious increase in the size and weight of the spectrometer, which limits its applications in aerospace. Consequently, many international research institutions are devoted to research on immersed echelle grating [21]. The concept of immersed gratings was first proposed by Hulthén in 1954 and published in Nature [22]. Szumski et al. systematically explored the characteristics of immersion gratings and explained the principle of using immersion beams to improve spectral resolution, luminous flux of immersed gratings under blazed conditions, and ghost image problems [23]. The efficiency, dispersion, and stray light performance of immersion gratings were tested by Cugny et al., and the findings were used to compare this method to the analytical model [24]. On the basis of analyzing the diffraction angle and angular dispersion of immersion gratings, Tang Qian et al. comprehensively discussed the dispersion nonlinearity of immersion gratings, showing that the spectral resolution of immersion gratings changed with change in the refractive index [25,26,27,28]. The development of immersion grating processing and manufacturing technology in European and American countries is relatively advanced. The main technologies are dominated by Canon USA [29,30], Lawrence Livermore National Laboratory [31], and the University of Texas (UT).

The UT mainly produces high-quality immersion gratings, with gratings suitable for near-infrared (1.1–5 µm), as well as mid- and long-wave infrared (5–35 µm) [32], applications, which were used on the Harlan J. Smith Telescope at the McDonald Observatory. For the immersion grating in IGRINS, silicon was chosen as the immersion medium for observations of H-band and K-band atmospheric windows in the study of young stars and protoplanetary systems [33]. Canon USA is committed to researching immersion grating processing and manufacturing technology. For example, common infrared crystals such as CdZnTe, Ge, and InP have been successfully used in immersion grating materials, and they can be designed in the wavelength range of 1.5–20 μm spectrometers with immersion gratings [34,35,36]. The Lawrence Livermore National Laboratory, with support from the National Department of Energy, has used immersed gratings in spectroscopic instruments for wastewater monitoring [37,38]. The National Aeronautics and Space Administration (NASA) developed a shell infrared high-resolution spectrometer for the astronomical telescope in Mauna Kea, Hawaii, which achieves ultra-high resolution through silicon immersion gratings, for the study of molecular absorption lines in astronomical observations [39,40]. In addition to the above-mentioned main research institutions, the research group of Professor Shen Weimin of Soochow University, has conducted in-depth research on the processing and application of silicon immersion gratings and designed a symmetric trapezoidal-groove silicon immersion grating. Subsequently, they developed a compact high-detection-accuracy 2 ppm atmospheric CO_2_ column concentration detection imaging spectrometer optical system [41,42]. To detect the target infrared radiation, a spectrometer must have a high spectral resolution. Using common grating, the instrument’s volume increases with the spectral resolution, which is not conducive for remote-sensing detection on airborne platforms. To solve this problem, this study uses a high-refractive-index crystal as the immersion material. The immersion grating is used as the main dispersion element of the spectrometer, where diffraction occurs in the medium. The dispersion and spectral resolution are improved by *n* times compared to traditional grating.

Depending on the characteristics of the infrared radiation of a detected target, a spectrometer’s optical system index parameters are determined. A high-resolution mid-infrared cooled echelle spectrometer is designed to solve the problem of increased volume caused by traditional grating, and a dispersion model of the immersion beam spectrometer combined with the theory of echelle grating is given. Zemax optical design software is used for simulation, optimization, and analysis of the system, and the design objective of real-time detection of the 3.7–4.8 μm infrared medium-wave spectrum is reached. The spectrometer designed in this paper has a simple structure and presents a new design idea for research on spectral detection in the middle infrared band.

The first section takes the example of a rocket tail jet and analyzes the characteristics of the infrared radiation of the target according to the computation of the flow field. The second section gives the theoretical basis of immersion grating, explains the dispersion and spectral resolution of immersion grating, analyzes the aberration characteristics of toroidal uniform line grating, and shows that as a secondary beam-splitting device compared with advantages of other devices, such as prisms, plane gratings, etc. The third section introduces the design method of the optical system, including the spatial layout of the cross-dispersion optical system, the design of the infrared lens, and the matching of the cold aperture of the cooled infrared detector. The fourth section evaluates the design results of the optical system design, including the diffraction characteristics of the echelle grating, the athermalization of the infrared lens, and the evaluation of the spectrometer system.

## 2. Materials and Methods

### 2.1. Infrared Radiation Characteristics of Rocket Plume

It has become a new research method to study effective countermeasures by building an infrared-guided missile simulation model. Research into the characteristics of a target’s infrared radiation is the foundation for the simulation of infrared-guided missiles. The calculation of the tail flame flow field is used to determine the infrared radiation of the tail flame. The primary purpose is to obtain the tail flame’s temperature and composition distribution data through the flow field calculation. The distinguishing features of a rocket’s tail plume are turbulence and re-ignition. The gas ejected from the nozzle is launched in the form of a series of waves, slowed down by the viscous resistance of the air, and finally reaches equilibrium with the outside gas. Thus, a rocket plume model needs to account for complex urban chemistry and gas-solid interactions. Figure 1 shows the infrared radiation of a missile plume.

Regarding the division of the infrared band range, different majors, such as the International Commission on Illumination, the International Astronomical Society, and the Optical Communication Division, have different division methods according to their own application needs. At present, the more commonly used division methods take into account the atmospheric window and infrared detectors and divide the infrared band into five bands: near-infrared (NIR), short-wave infrared (SIR), mid-wave infrared (MIR), long-wave infrared (LIR), and far-infrared (FIR). As shown in Figure 2, according to the infrared radiation characteristics of a ballistic missile’s plume during the active segment’s flight, the radiation intensities of the plume at different wavelengths are different, so it is necessary to carry out space-based early warning for various infrared bands. The missile propellant fuel varies with the type of missile, mainly including solid fuel, liquid fuel, and solid-liquid mixed fuel. Still, no matter what energy is used, the product’s main components after combustion are H_2_O and CO_2_, as well as the molecules of these two gases. The structure determines that the infrared radiation energy of the missile tail flame is more substantial in the 4.3μm and 4.8μm bands. At the same time, consider the atmospheric window and the parameters of the infrared detector. To determine one of the important parameters of the spectrometer, the band range is 3.7–4.8μm.

### 2.2. Immersion Grating Theory

The spectroscopic principle of grating is essentially multi-slit Fraunhofer diffraction, and the most commonly used is plane-reflection echelle grating. As shown in Figure 1, according to the principle of multi-beam interference, the dispersion equation of plane echelle grating is:(1)d(sinα±sinβ)=mλ

Among them, *d* is the grating constant, *α* is the grating incident angle, *β* is the diffraction angle, *λ* is the incident wavelength, and *m* is the diffraction order of *m* = 0, ±1, ±2, … ±n. An immersion grating is a diffractive optical element that combines diffraction grating with a prism (immersion medium), where the directive surface is immersed in a material with a high refraction index, resulting in an improved resolution compared to a non-immersion grating of the same size. For an increase in times (*n* is the refractive index of the medium), the maximum spectral resolution of immersion grating at different wavelengths is illustrated in Figure 3 [43,44,45].

As shown in Figure 3, the light passes through the incident surface of the grating and then passes through the diffraction immersion grating equation. As shown in Equation (1), the off-plane angle of the grating is as follows:(2)mλ=ndcosγ(sinα+sinβ)

The equations show that the main difference between immersion and planar reflective echelle gratings is the presence or absence of a high-refractive-index immersion medium. As shown in Figure 4, when light with a wavelength λ enters a prism, the effective wavelength $\lambda_{e}$ that reaches the diffraction surface of the grating is *n* times smaller, that is, the refractive index of the prism is defined as $\lambda_{e}=n\lambda$. Therefore, the wavelength-to-grating-constant ratio $\lambda_{e}/d$ is also reduced.$\;\lambda_{e}/d$ determines the overall photometric efficiency of the stepped-prism combination. In addition, it can be understood that the effective grating constant of the echelle is increased *n* times, and because the number of ruled lines remains unchanged, this also leads to an increase in the effective length of the echelle grating by a factor of *n*; the spectral resolution increases *n* times, and the dispersion power is reduced by *n* times. Under the Littrow condition of α=β and differentiated for the wavelength λ, the dispersion of the immersion grating consists of two parts: the refractive index and the diffraction angle. The angular dispersion of grating can be written as:(3)$$\;\frac{nd\cos\beta }{m}d\beta =d\lambda$$
dλ/αβ is negligible (Δβ), so:(4)$$\frac{nd\cos\beta }{m}\Delta \beta =\Delta \lambda$$

The spectral resolving power *R*, thus, becomes:(5)$$R=\frac{\lambda }{\Delta \lambda }=\frac{\lambda }{nd\cos\beta \Delta \beta }=\frac{\sin\alpha +\sin\beta }{\cos\beta \Delta \beta }$$

**Figure 4 sensors-22-07291-f004:**
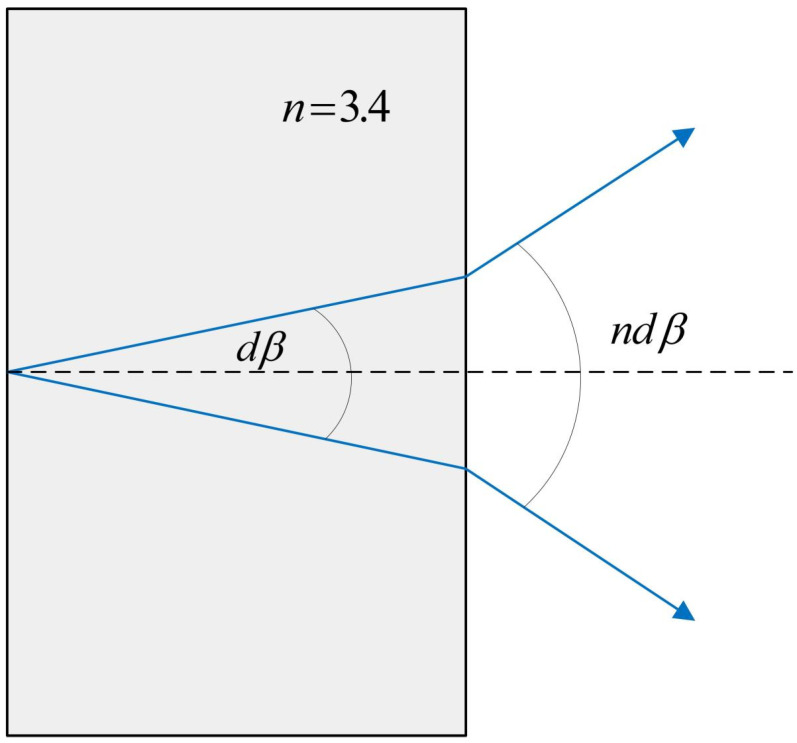
Increase in dispersion on the exiting immersion.

Immersion gratings of infrared materials such as silicon and germanium have been successfully applied to satellite and ground-based high-spectral-resolution spectrometers. Because single-crystal silicon is easy to process in the middle-infrared range and has a relatively mature processing technology, this study used single-crystal silicon as an immersion material. Some semiconductors have potential in the infrared band as a high-refractive-index material. A single-crystal silicon transmits light at 1.2–7.0 µm and can be applied to immersion echelle gratings by photolithography. The single-point diamond machining of softer materials, including CdZnTe, is a method for producing echelle immersion grating with wavelength bandwidths different from those of silicon crystals.

Figure 5 shows attenuation coefficients for the major candidate materials. The data are cited and plotted from [46] and [47], except for the InP data, which were recently measured using the same method. The right scale of the vertical axis shows the energy loss in the case of an immersed grating with a basic length of 80 mm [29]. Silicon has a wide range of applications in modern technology, such as transistors, diodes, and solar cells. In astronomy, however, Si is the basis for silicon immersion gratings, an important and useful optical material because of its optical and mechanical properties. Additionally, process technologies for semiconductor VLSI electronics and MEMS applications have been developed. Although there are a few surgeons that silicon absorbs, high-purity silicon transmits well in near- and middle-infrared wavelength regions (1.2–40 µm) [48,49,50] and above. Over the last decade, a number of research groups have developed methods for manufacturing diffraction gratings on silicon substrates [51,52,53].

### 2.3. Dispersion Characteristics of Toroidal Gratings

Toroidal gratings can correct astigmatism through different curvature radii in two directions [54]. Compared to spherical gratings, toroidal gratings are generally able to eliminate astigmatism at two points in the spectrum. By adjusting the sizes of the angle of incidence and the angle of diffraction, these two astigmatic points can be tuned to any desired wavelength over a wide range of the spectrum.

A detailed discussion of the optical path difference function for toroidal gratings can be found in [55]. In actual design, it is impossible to completely satisfy the ideal imaging conditions, and $F_{\mathrm{ij}}=0$ can only be made close to 0 to minimize the aberration. We express the conditions for spectraland spatial on-axis focusing, respectively, with:(6)$$\begin{array}{cc} \frac{\cos^{2}\alpha }{r_{A}}+\frac{\cos^{2}\beta }{r_{B}}-\frac{\cos\alpha +\cos\beta }{\rho \;}=0 & \\ & \mathrm{horizontal}\;\mathrm{focal}\;\mathrm{curve} \end{array}$$
(7)$$\begin{array}{cc} \frac{1}{r_{A}}+\frac{1}{r_{B}}-\frac{\cos\alpha +\cos\beta }{R\;}=0 & \\ & \mathrm{vertical}\;\mathrm{focal}\;\mathrm{curve} \end{array}$$

The schematic diagram of concave grating is shown in Figure 6. The above formula is the focusing condition of the concave grating meridian beam. To obtain a clear and sharp spectral line, the relative positions of the incident slit, the concave grating, and the image of the slit must satisfy the above fundamental relationship. Toroidal gratings can correct astigmatism through different curvature radii in two directions. First, the radii of curvature in the sagittal and meridional directions of the toroidal grating are determined. Then, the meridional focal length of the toroidal grating used to correct for astigmatism is calculated. Finally, the calculated parameters are input into the optical design software Zemax for ray tracing and optimization to verify the performance of the optical path system.

From the horizontal focusing Equation (6), the incident arm r and the exit arm r’ are obtained as:(8)$$r_{A}=\rho \cos\alpha \;r_{B}=\rho \cos\beta$$

Combining Equation (8) and vertical focusing condition Equation (7), the astigmatism correction conditions are as follows:(9)R/ρ=cosαcosβ

It can be known from grating Equation (1) that the diffraction angle β varies with the wavelength λ under the condition of a certain incident angle α. The designed wavelength range is $\lambda_{1}$~$\lambda_{2}$, and the center wavelength is $\lambda_{c}=(\lambda_{1}$~$\lambda_{2})$/2. In order to correct astigmatism at the same time in a wide band, the diffracted light of the center wavelength is designed to be along the normal direction of the grating, that is:(10)$$\sin\alpha =\frac{m\lambda_{c}}{\sigma_{0}}$$

The diffracted rays of edge wavelengths $\;\lambda_{1}$ and $\lambda_{2}$ are located on both sides of the grating normal, and $\beta (\lambda_{1}$) = $\beta (\lambda_{2}$).

The incident angle α and the grating groove density $\sigma_{0}$ are selected according to the band range and the center wavelength, and then the diffraction angle β is calculated corresponding to different wavelengths according to formula (7), and the horizontal radius ρ is determined according to the spectral resolution requirements and the grating groove density $\sigma_{0}$. Finally, the vertical radius *R* is calculated according to Formula (18). Firstly, the initial optical structure parameters are solved according to Equations (6)–(10), and then the initial structure parameters are optimized by Zemax software.

## 3. Optical System Design Method

A reflection grating spectrometer is not limited by materials, and there is no chromatic aberration. There are four types of optical systems for classical reflection grating spectrometers: Czerny-Turner, C-T, Ebert-Fastie, and Littrow. Their institutions are shown in Figure 7, and Table 1 presents the advantages and disadvantages of each optical path.

The Czerny-Turner optical path is one of the most widely used spectrometer structures for one-dimensional medium- and high-resolution spectrometer structures owing to its compact structure, non-moving parts, and flat image field characteristics [56,57]. The echelle grating spectrometer utilizes an echelle grating and a low-dispersion lateral light-splitting element to form a two-dimensional overlapping spectrum on the image plane. The low-dispersion element can choose grating or a prism.

Usually, the composition of a dispersion system is roughly divided into two types. The first main dispersion element is a grating, and the second-order beam-splitting element is a prism. In this method, the prism can have a higher optical efficiency, no blaze, and order overlap, but the disadvantage is that the dispersion is not uniform, and the working wavelength range is limited by materials. The second type uses two gratings, one as the main dispersion element and the other as the second-order beam-splitting element. Using a grating as the second-order beam-splitting element can make the dispersion more uniform and the working wavelength range wider [58,59]. As shown in Figure 7a, when working, the light entering from the incident slit first passes through a collimator mirror to become quasi-parallel light, and then it is dispersed by the grating and imaged at the exit slit by the focusing mirror. The optical system of a spectrometer with a Czerny-Turner structure has great flexibility, and it can be roughly divided into M-type and Z-type in its structure.

The design an optical system takes considers the overall resolution of the instrument and the volume of the whole system. To improve and streamline the optical structure of an echelle spectrometer, the Czerny-Turner and quasi-Littrow structures were used for cross-designing. In this study, a spectral detection technology principle based on TULS+ immersion echelle grating was proposed, as shown in Figure 8. The optical system consisted of (1) a spectrometer slit, (2) a folding mirror, (3) a collimating mirror, (4) an immersion echelle grating, (5) a cross disperser, (6) an imaging mirror, (7) a cold aperture, (8) a detector, etc. To obtain a higher spectral resolution, grating spectrometers often need to use higher-order diffracted lights. Secondary dispersion optics must be added to the optical path to prevent spectral aliasing due to the free spectral range. A concave toroidal grating with uniform line spacing was proposed. The grating base was a toroidal surface, and the grating line spacing was uniform. There were two highlights in optical design that need to be introduced. First, the traditional plane grating was replaced by immersion grating to achieve miniaturization and higher light intensity. Secondly, as a subdispersive element, toroidal grating has the following advantages: toroidal grating horizontal dispersion is different from vertical dispersion, which can also realize the separation of orders and the convergence of light beams, simultaneously meeting the requirements of high energy utilization and high imaging quality.

Cooled infrared optics are perfect for high-sensitivity and high-resolution infrared detection. Currently, an echelle grating spectrometer optical system is mainly used in visible light and uncooked infrared fields, and it is less-used in cooled infrared spectrometers. Due to the need to match the cold diaphragm of the cooled infrared camera, its optical system needs to set an exit pupil or aperture diaphragm between the image plane and the last optical surface to effectively avoid background radiation intensity outside the field of view entering the image through the cold window. As shown in Figure 9, to ensure that the background radiation outside the field of view is not received from the image plane, a 100% cold-stop effect must be achieved in the optical system.

Few infrared materials transmit these wavelengths and have poor thermal properties. Commonly used IR materials are Si, Ge, ZnSe, and ZnS, the thermal properties of which are shown in Table 2. The formalization design could generally be divided into three steps: The first step was to find a suitable initial structure at room temperature (20 °C) and optimize it to obtain a room-temperature system with a better image quality. The second step, based on increasing the temperature expansion coefficient of the lens barrel material, analyzed the image quality at each temperature point and checked the defocus amount at each temperature point. If the image quality and defocus amount were relatively poor, the operator must go back to the first step. In the third step, the defocus amount range of high- and low-temperature points was controlled, and temperature compensation was optimized and performed at each temperature point until a better image quality was achieved. The material of the lens barrel was aluminum, and its linear expansion coefficient was 23.6 × 10^−6^.

The performance indicators of the portable spectrophotometer designed in this paper were based on the indicators of more-advanced foreign instruments of the same kind and fully considered design, research, and development costs. For the calculation of parameters, the relationships among the parameters were obtained, as follows in Table 3. It is emphasized that the toroidal grating could be easily fabricated by mechanical scribing on the toroidal substrate.

Figure 10 shows the relationships between the spectrometer design parameters and detection targets.

## 4. Results and Discussion

### 4.1. Analysis of Echelle Grating Characteristics

The physical properties of echelle gratings are usually represented by the tangent value *R* of the blaze angle (R=tanθ) and the density of the grating lines. The blaze angle of echelle gratings is typically 63.4° or 75°. Blaze angles of 63.4° and 75° are often referred to as “R2 gratings” and “R4 gratings”, representing tangents of R=2 and R=4 for 63.4° and 75°, respectively. According to the dispersion formula of the grating, the angular dispersion of R4 grating is twice that of R2 grating. The resolution of an echelle grating is related to the total effective reticle width W, diffraction angle *θ*, deflection angle *γ*, and wavelength *λ*. For a particular echelle grating, the resolution can be calculated according to Equation (5), where the wavelength is known as the diffraction angle; therefore, the resolution of an echelle grating can be determined using the working parameters of the grating (the width of the grating, the deflection angle, and the wavelength). Figure 11 shows the relationships between grating line density and diffraction order for different immersion materials and blaze angles.

Through the above analysis of the diffraction model of echelle grating, the primary conditions for the operation of the echelle grating could be obtained. Here, an echelle grating with a blaze angle of 63.4°, a 75° line density of 31.6 lines/mm, and 79 lines/mm was taken as an example. Assuming an incident wavelength of 200–1200 nm, Figure 12 shows what the spectral formats of the R2 and R4 gratings looked like for a given spectral resolution, according to Equation Varies with reticle density. The left subpictures (a) and (c) of Figure 12 correspond to 31.6 lines/mm and 79 lines/mm, respectively, corresponding to R2 grating. The subfigures (b) and (d) on the right side of Figure 12 correspond to R4 gratings of 31.6 lines/mm and 79 lines/mm, respectively.

Because the working band of the echelle grating spectrometer was selected as 3.7–4.8 μm, the diffraction order corresponding to each reference wavelength could be calculated according to the grating equation of the off-axis plane. The optical system was designed with multiple structures, and the diffraction orders of the grating were selected as m = 120, 127, 135, 147, and 155 for the calculations. According to the calculations of formula 1 for the central wavelength of each order and formula 2 for the free spectral range, the diffraction order range corresponding to the working band of 3.7–4.8 μm could be calculated to be 120 to 155 orders. Table 4 lists the centers of these orders for wavelength, limit wavelength, and free spectral range. For each sampled value of m, the selected wavelength data were as follows.

From Figure 13, the entire wavelength range was composed of several diffraction orders, and each order corresponded to a certain width of the free spectral region. The shorter the wavelength was, the narrower the width of the free spectral region, that is, the change in the diffraction angle. The smaller the range, the larger the angular dispersion; the longer the wavelength was, the wider the width of the free spectral region, that is, the larger the variation range of the diffraction angle, but the smaller the angular dispersion.

### 4.2. Thermal Design for Infrared Lens

In the mid-infrared band, unlike the visible and near-infrared bands, the background noise generated by thermal radiation in this band increases significantly, the signal intensity from the sun drops sharply, and the thermal radiation and stray light from the optical system components and supporting structures in the spectrometer reach the image plane, which is imaged together with the target beam and becomes the main source of noise.

The temperature of the application environment of an infrared echelle spectrometer varies greatly. Therefore, the infrared-imaging lens needs to add a temperature compensation link to ensure that the infrared system is always in the best imaging state, which means thermal properties of the infrared-imaging system need to be considered during the designing stage. In the formalized design, it was necessary to consider the changes in the refractive index, radius, and thickness of the infrared lens element with temperature, as well as the influence of thermal expansion and contraction of the lens barrel material with temperature. The cumulative effect of the temperature changes in lens elements and barrel materials on the infrared-imaging system is that the focal plane of the infrared system was displaced, making the image unclear. The purpose of the formalized design was to maintain the focal plane offset in a single focal depth. According to the Rayleigh criterion, when the displacement of the focal plane is less than the focal depth of a system, the maximum wave aberration between the actual wavefront and the reference wavefront does not exceed one-fourth of the wavelength. This wave front can be considered defect-free and can achieve a relatively ideal imaging effect, depending on the purpose of the formalized design. Figure 14a–d is the image quality evaluation diagram of the infrared lens without athermalization design. Figure 15 shows the aberration curve of the infrared lens at 20 °C.

Figure 16a–c shows the meridional and sagittal MTF curves of each field of view of the formalized optical system at ambient temperatures of 20 °C, −50 °C, and 50 °C and a Nyquist frequency of 16 l/mm. It is shown that that the meridional and sagittal MTF values of each field of view were all greater than 0.6, and the MTF values at the Nyquist frequencies of the 0.7 fields of view were all greater than 0.5, which was close to the diffraction limit. Table 5 shows the defocus values of infrared lens after athermalization at different three temperatures.

Figure 17 is a 3D image of the optical system simulation of the echelle spectrometer. The following describes the constraints of the parameters of the spectrometer components. The spectrometer design should meet the detection requirements in the spectrum range of 3.7~4.8 μm, and optional detector parameters should also be considered. Based on the type of spectral line broadening of the gas in the infrared band, to meet the above requirements, the spectral wavenumber resolution should be at least 0.1 cm^−1^, and the corresponding spectral resolution should be in the order of 120–155. The spectrometer system was designed to use a domestic mercury cadmium Telluride (HgCdTe) focal plane array device with a pixel size of 30 μm × 30 μm and an array size of 320 × 256. Based on the relationship between the spectral resolution and the size of the detector image plane, the spectrometer grating adopted a custom immersed echelle grating: the grating number was 12.36 lines/mm, the blaze angle was 63.4°, and the immersion material was a single-crystal silicon.

As shown in Figure 18, we investigated the dispersion properties of the immersion grating. We found that, when immersion gratings were adopted, significant differences were observed in the distribution of spectral lines compared with normal gratings, and the echellogram spectral lines tilted [60]. From the figure, the maximum length occupied by the image plane was 9.6 × 7.68 mm, which was smaller than the effective photosensitive area of the detector and could be detected with the se-lected detector.

As shown in Figure 19 and Table 6, the spot diagram of the center wavelength of diffraction orders in the spectral range of 3.7~4.8 μm was analyzed, and the RMS size of the light spot. In Figure 19a–i, each column is the same order and different wavelengths; each row is different orders. In the figure, it is shown that the designed optical system of the echelle spectrometer had a large change in the RMS size of the central wavelength spot diagram in the short-wave range, resulting in a relatively poor spectral resolution in the short-wave range. For the long-wave range, although the RMS size of the central wavelength spot diagram was small and it did not change much, its spectral resolution was relatively poor due to the relatively small linear dispersion of the long wave. However, the spectral resolution of the designed mid-step spectrometer at the center wavelength of each diffraction order met the 0.1 nm spectral resolution required for the design.

## 5. Conclusions

To fulfill the demand for the remote sensing of tail flames, this study proposed a new broadband, high-resolution, infrared spectrum detection system based on the diffraction characteristics of toroidal uniform gratings and the theoretical basis of immersed echelle gratings. The spectrometer had a working band of 3.7~4.8 μm and a central wavelength of 4.25 μm. The spectrometer utilized a 320 × 256 focal plane array detector as the receiver and an immersed echelle grating as the spectroscopic element. To reduce the infrared background radiation, the spectrometer adopted a cooling infrared detector; its arrangement adopted a combination of the Littrow and Czerny–Turner structures to compress the optical path and correct aberrations. The theory of toroidal grating was applied to the design of the echelle spectrometer to form a broadband, compact infrared spectrometer. The optical design software Zemax was used for ray tracing, and the design results were analyzed. The results showed that the design fully met the requirements of imaging quality, and the structure was compact, making full use of the toroidal grating’s characteristics of dispersion, focusing, and correction of aberrations.

Consequently, compared with the method of folding the optical path once, the volume of the vacuum refrigeration system could be reduced effectively while satisfying the image quality. This solution effectively solved the contradiction between broad spectral range and high spectral resolution, as well as the problem of having a large grating size due to a high spectral resolution. This system had a comprehensive spectral coverage, ultra-high spectral resolution, high detection sensitivity, high repeatability, and on-demand spectroscopy. Therefore, it can be implemented on various platforms, such as spaceborne and airborne objects. Additionally, it can be developed into a general scientific instrument that has broad application prospects in the fields of atmospheric detection and environmental monitoring. This study lays a good foundation for the application and further development of this new detection technology in infrared remote sensing and other fields.

## Figures and Tables

**Figure 1 sensors-22-07291-f001:**
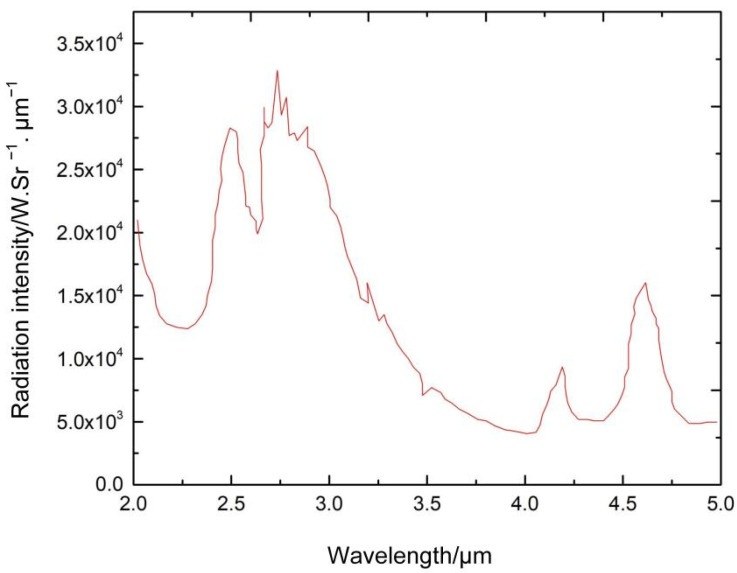
Spectral radiant intensity of a plume.

**Figure 2 sensors-22-07291-f002:**
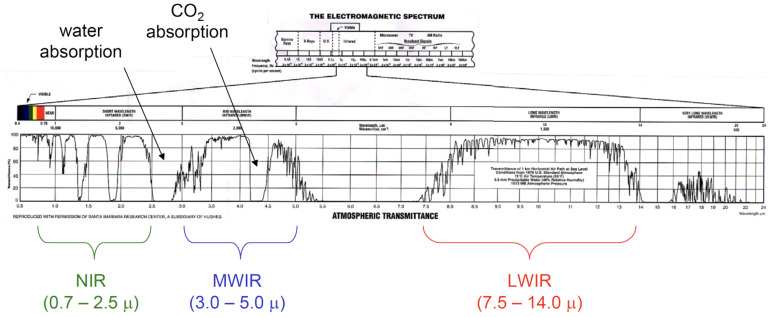
Infrared band atmospheric transmittance.

**Figure 3 sensors-22-07291-f003:**
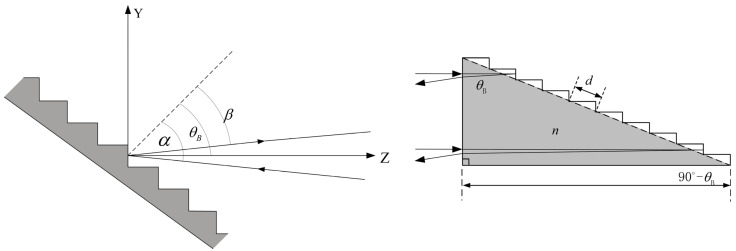
Conventional reflective grating and immersion grating.

**Figure 5 sensors-22-07291-f005:**
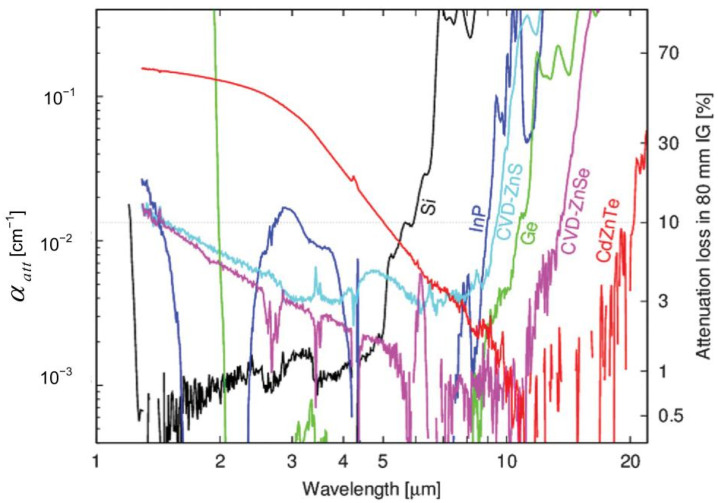
Infrared material properties of candidate infrared materials.

**Figure 6 sensors-22-07291-f006:**
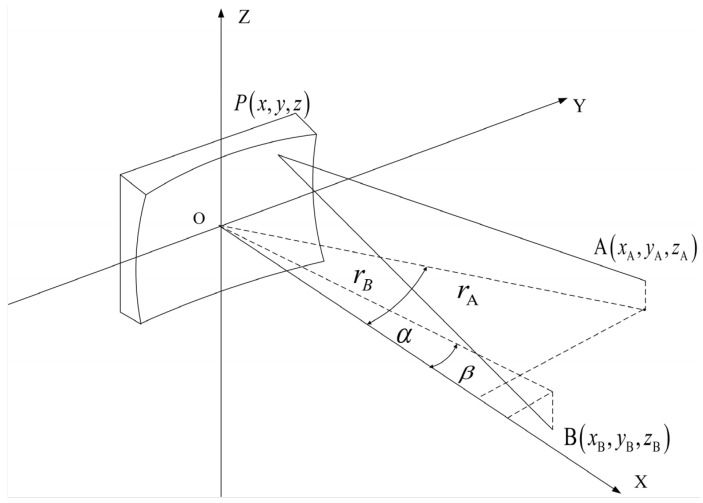
Schematic of the optical layout of concave grating.

**Figure 7 sensors-22-07291-f007:**
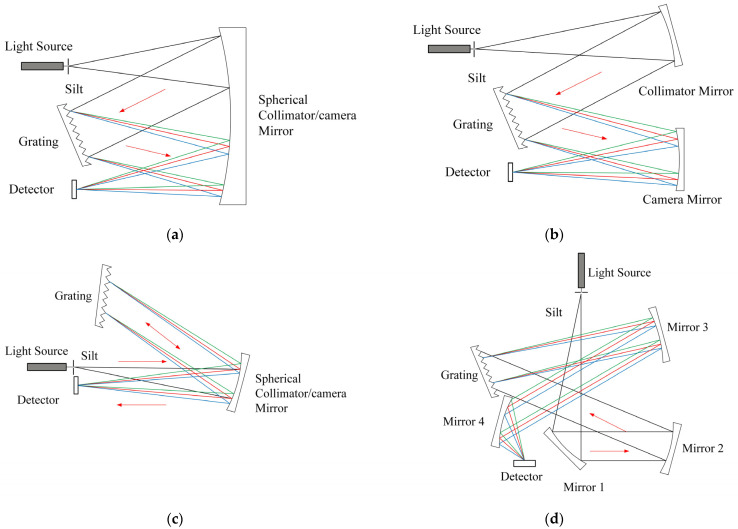
Optical structures of typical plane grating spectrographs:(**a**) Ebert-Fastie; (**b**) Czerny-Turner; (**c**) Littrow; and (**d**) Chupp-Gtantz.

**Figure 8 sensors-22-07291-f008:**
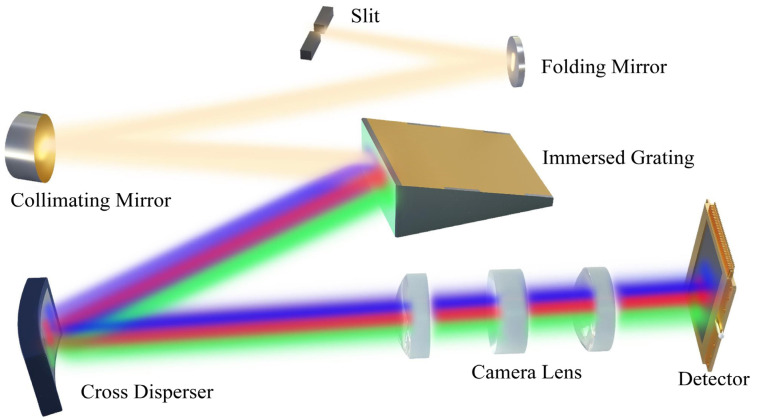
3D schematic of cross-dispersion structure of echelle spectrometer.

**Figure 9 sensors-22-07291-f009:**
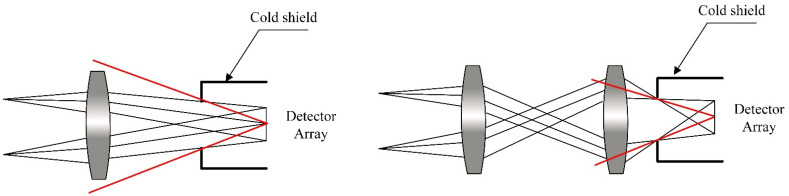
Schematic diagram of cold-stop matching of infrared optical system.

**Figure 10 sensors-22-07291-f010:**
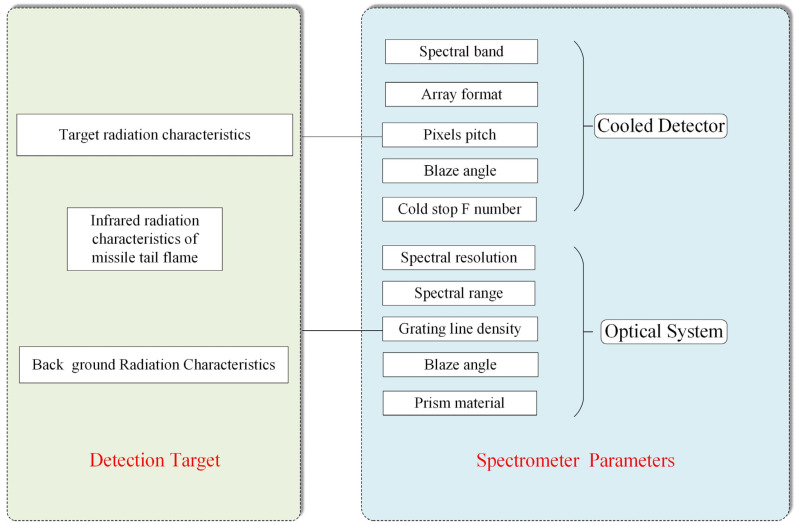
The relationships between spectrometer design parameters and detection targets.

**Figure 11 sensors-22-07291-f011:**
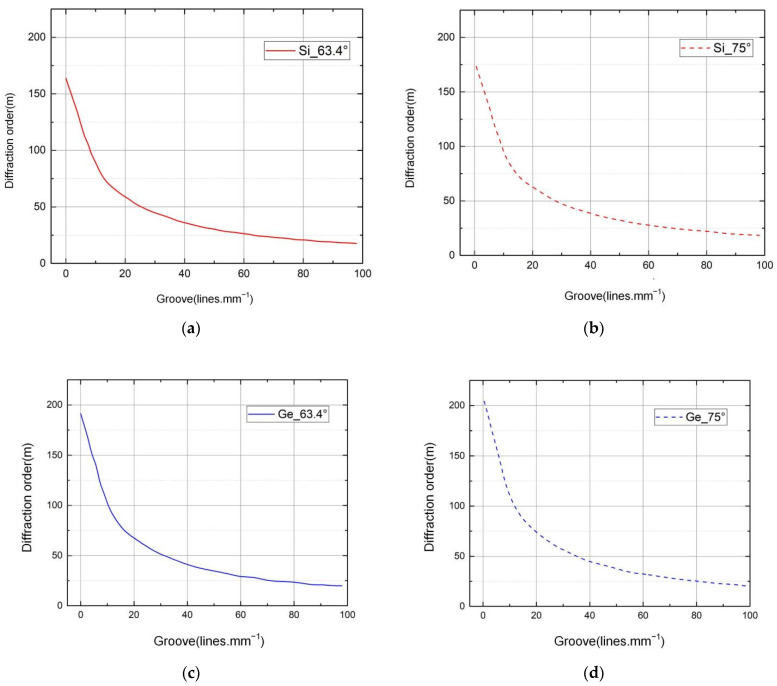
Relationships between grating line density and diffraction order for different immersion materials and blaze angles: (**a**)silicon immersion grating, blaze angle 63.4°; (**b**) silicon immersion grating, blaze angle 75°; (**c**) germanium immersion grating, blaze angle 63.4°; (**d**) germanium immersion grating, blaze angle 75°.

**Figure 12 sensors-22-07291-f012:**
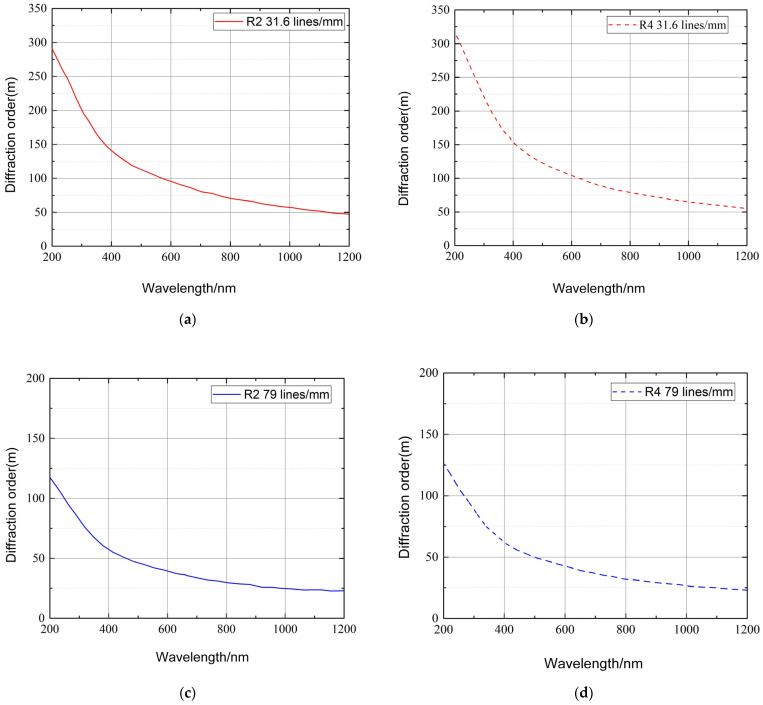
Relationships between wavelength and diffraction order for typical grating line densities and blaze angles: (**a**) 31.6 lines/mm, blaze angle 63.4°; (**b**) 31.6 lines/mm, blaze angle 75°; (**c**) 79 lines/mm, blaze angle 63.4; (**d**) 79 lines/mm, blaze angle 75°.

**Figure 13 sensors-22-07291-f013:**
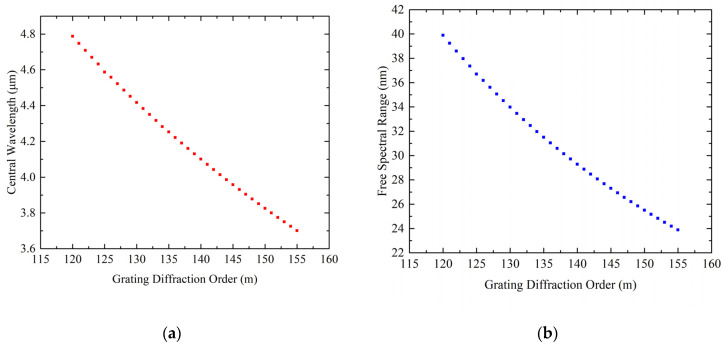
Plots of central wavelength, free spectral range, and diffraction order: (**a**) diffraction order versus center wavelength; (**b**) diffraction order versus free spectral range.

**Figure 14 sensors-22-07291-f014:**
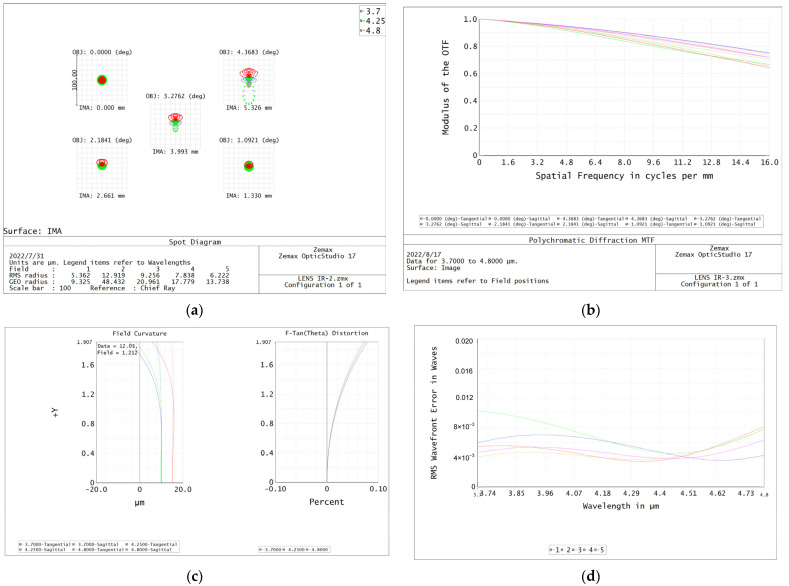
Infrared lens image quality evaluation: (**a**) spot diagram; (**b**) MTF curves; (**c**) field curvature and distortion curves; (**d**) RMS wavefront error versus wavelength.

**Figure 15 sensors-22-07291-f015:**
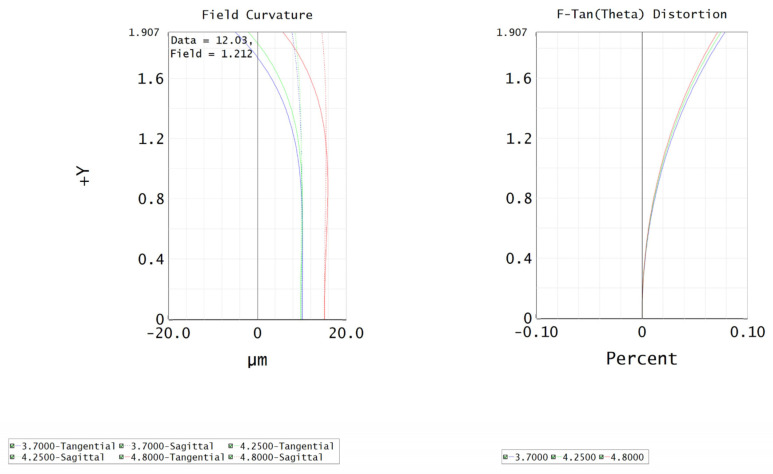
Aberration curve of infrared lens at 20 °C.

**Figure 16 sensors-22-07291-f016:**
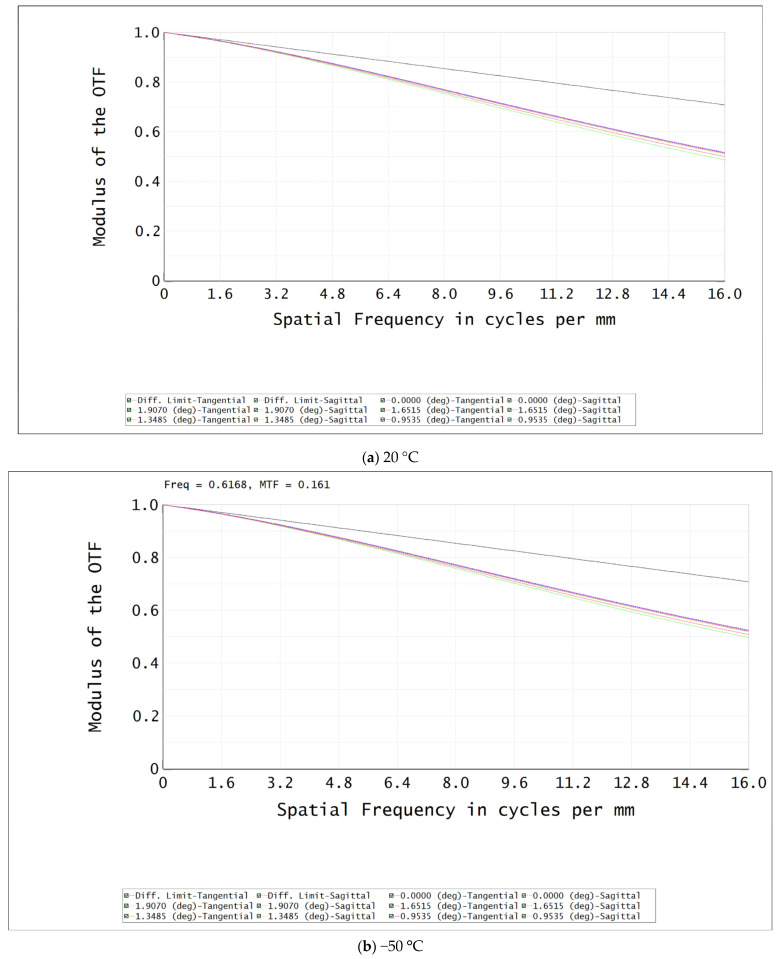
Plots of MTF curves of optical system at different temperatures: (**a**) 20 °C; (**b**) −50 °C; (**c**) 50 °C.

**Figure 17 sensors-22-07291-f017:**
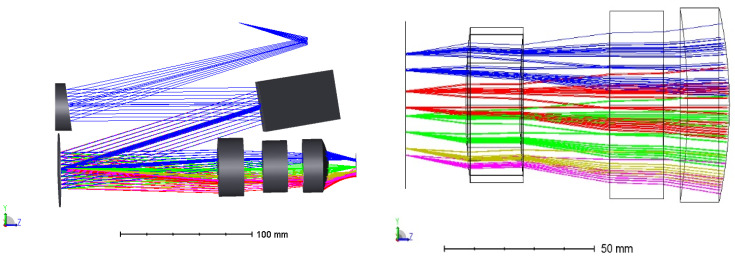
Optical layout of MIR-cooled spectrometer.

**Figure 18 sensors-22-07291-f018:**
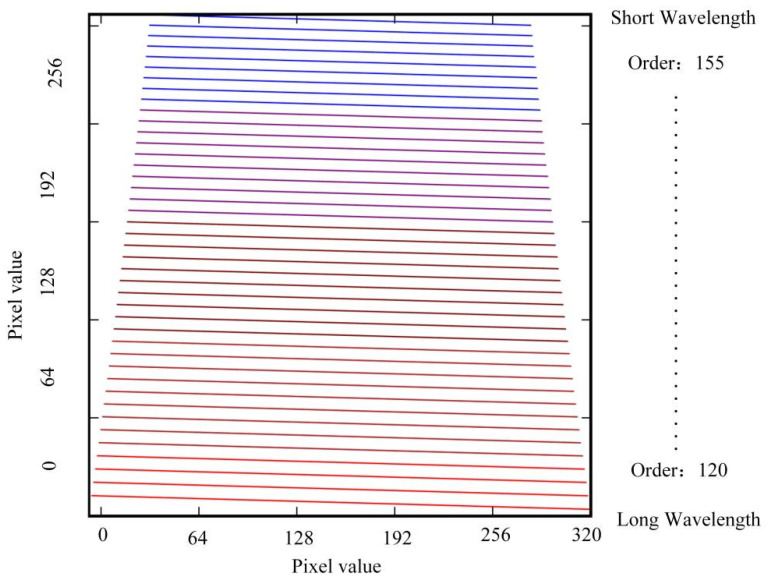
The echellogram shows the free spectral range of 35 orders on a detector of 320 × 256 pixels.

**Figure 19 sensors-22-07291-f019:**
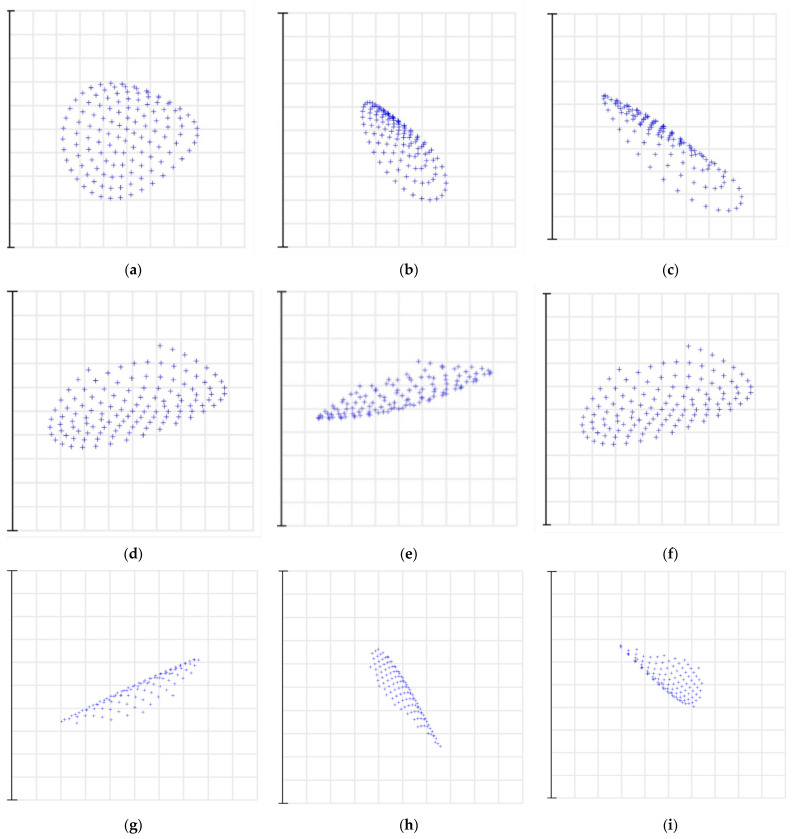
Spot diagrams corresponding to different diffraction orders.

**Table 1 sensors-22-07291-t001:** Comparison of advantages and disadvantages of various optical paths.

Optical Layout	Advantage	Disadvantage
Figure 7a Ebert-Fastie	Structure is symmetrical Small residual coma Compact Convenient adjustment	Large volume size Secondary diffraction Multiple diffractions Stray light
Figure 7b Czerny-Turner	Avoid multiple diffractions Beneficial to adjustment	None reported
Figure 7c Littrow	Simple structure Compact structure	Multiple diffractions High stray light Spectral smile and keystone
Figure 7d Chupp-Gtantz	High resolution Low stray light	OAP difficult process

**Table 2 sensors-22-07291-t002:** Optical and thermal properties of infrared materials.

Material	Wavelength/μm	Refractive Index	CTE/×10^−6^
SiO_2_	0–4.5	1.48	4–10
MgF_2_	0.45–9.5	1.34	11.5
MgO	0.4–10	1.7	13.9
CaF_2_	0.2–12	1.37	20
ZnS	0.6–15	2.2	7
ZnSe	0.5–22	2.4	7.7
Si	1.3–15	3.42	4.2
Ge	1.8–25	4.02	6.1

**Table 3 sensors-22-07291-t003:** The parameters designed for the optic components.

Parameters	Values
Wavelength coverage	~3.7–4.8 μm
Resolving power R	30,000
Entrance F number	8
Parabolic mirror	Reflected focal length: 275 mm
Main disperser	Immersion grating blaze angle: 63.5° (R2)
Cross disperser	Toroidal grating tangential radius: 500 mm; sagittal radius: 484.5 mm
Detectors	Array format: 320 × 256 Pixel size: 30 μm × 30 μm

**Table 4 sensors-22-07291-t004:** The wavelengths and free spectral ranges of multiple configurations.

Configuration	1	2	3	4	5
m	120th	127th	135th	147th	155th
$\lambda_{\mathrm{cen}}/\mu m$	4.79	4.53	4.26	3.91	3.70
$\lambda_{\min}/\mu m$	4.77	4.51	4.24	3.90	3.70
$\lambda_{\max}/\mu m$	4.81	4.54	4.27	3.92	3.71
FSR/nm	39.93	35.65	31.55	26.61	23.93

**Table 5 sensors-22-07291-t005:** Defocus values of the optical system.

Temperature/°C	50	20	0	−20	−50
Defocus/μm	14.2	12.1	0	6.3	9.0
Depth of Focal/μm	40				

**Table 6 sensors-22-07291-t006:** RMS and spectral resolution of multiple configurations.

Wavelength	B1	B2	B3	C1	C2	C3	T1	T2	T3
Figure	(a)	(b)	(c)	(d)	(e)	(f)	(g)	(h)	(i)
RMS/μm	25.2	32.3	25.8	32.8	39.7	32.5	39.8	25.6	26.2
Resolution	45,000	44,600	45,800	24,800	25,300	27,600	22,000	23,000	24,500

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
