# Peer review of "Design of Compact Mid-Infrared Cooled Echelle Spectrometer Based on Toroidal Uniform-Line-Spaced (TULS) Grating"

_sensors, 2022, doi:10.3390/s22197291_

Round 1
Reviewer 1 Report
The authors are intended to present the use of a toroidal grating for the realization of a compact MIR echelle spectrometer. Although the idea is nice, the presentation is quite disorganized, particuarly regarding a clear presentation of the advantage of the use of a toroidal grating.
In my opinion, the paper in the present form should be rejected, since it is required a substantial revision and reorganization of the theoretical presentation, design characteristics and result discussion.
In the following, some comments are listed.
Introduction.
Some of the general references are very old, going back to 1960 (ref.3) and 1967 (ref. 4). As introductory references, the authors may cite more recent textbook on optics and spectroscopy (easily found in the literature), since they are for sure more useful for the readers.
References 20, 22 and 23 on immersed gratings are also quite dated (to 20 years ago). Only one of these could be cited.
Optical path difference and related equations. What is the aim of presenting such complex formulas, that have been already extensively discussed in the literature? Text is not clear, saying only “Optical path difference” without any additional explanation. Formulas 6-11 can be omitted from the article, since they do not have any link with the following presentation.
Focusing equation for a toroidal grating
Why only one of the two focusing equations for a toroidal surface is listed? Since the paper is related to the use of a toroidal grating, focusing equations in both directions (tangential and sagittal) should be reported, where the effects of the two grating radii for 2D focusing have to be clearly presented. Please, upgrade consequently Eq. 12 adding the missing equation for the sagittal focus related to the choice of the sagittal radius.
At Page 7, authors cite also variable-space (variable-pitch) toroidal gratings, that are indeed not required in the presentation. Eventually, these types of gratings can be cited in the conclusions as possible alternate choice, although not practical for feasibility reasons.
Figure 6. “Collimating lens” is “collimating mirror”.
The optical system, as presented in the text describing Figure 6, introduces the “immersion echelle grating”, but the cross disperser is missing in the text. The quality of the description should be definitely increased.
Most important of all, the motivation to use a toroidal grating is not explained at all in the text. What is the driving parameter for the choice of the sagittal radius (that is obviously different from the tangential radius)? If this point is not clearly discussed, the main aim of the design (i.e., the use of a toroidal surface) is not clear to the reader.
Last but not least, converting Figure 6 from a 2D to a 3D schematic would make it much more clear.
In Table 3, the parameters of the toroidal grating are missed, that is, the two radii demonstrating that the surface is toroidal. The quantitative definition of the two radii is of primary importance, since the feasibility of the toroidal surface depends on the difference between tangential and sagittal radii.
The paper is related to the use of a toroidal grating. There is a full paragraph on “Thermal Design for Infrared Lens”, altough this item is not directly linked to the use of a toroidal grating. It is not clear to the reader why such a detailed study for the imaging lens is reported in the paper. The paragraph needs to be clearly introduced to the reader.
I hope that the suggestions may help the authors to revise substantially the presentation.
Author Response
Revised manuscript red comment.
Response 1: Please note that the ref.3 ref.4 numbering has changed to ref.4 、ref.5, due to the addition of new references to the previous text.
References [20], [22] and [23] on immersed gratings has been modified and replaced.
Response 2: Optical path difference and related equations. This section was not previously described and the latest manuscript has been revised and a detailed description of the optical path difference of toroidal gratings is given. At the same time, the advantages of toroidal gratings in spectrometer applications are also analyzed. Also formulas(6)-(11) have been revised. A detailed discussion of light-path function given in Ref[56].
Response 3:
1) The focusing equations in the meridional and sagittal directions of the toroidal grating are completed, and their effects on 2D focusing are analyzed. Refer to the revised manuscript for details.
2)At Page 7 of original manuscript , variable-space (variable-pitch) toroidal gratings has been deleted.
Response 4:
1)“Collimating lens”should be“collimating mirror”; I have checked and revised.
2) Added description of cross-dispersion to text of Figure 6 Optical system
3) According to the reviewer's comments, converted Figure 6 from a 2D to a 3D schematic.
Response 5:In Table 3, added the parameters of the toroidal grating of the two radii demonstrating .
Response 6: The necessity of the Thermal Design for Infrared Lens paragraph was introduced in the manuscript while also describing it more clearly to the readers.

Reviewer 2 Report
A high-quality mid-infrared hyperspectral imaging instrument is difficult to develop but has great potential for some target detection. The paper proposes a new mid-infrared spectrometer design based on the theoretical basis of echelle grating for recording the infrared spectrum. It has a scientific implication for the development of mid-infrared hyperspectral imagers. However, the paper should be improved on the following issues.
1. Line 42-43. Please add the reference.
2. Line 55. Please note the logical relationship between this sentence and the above sentences. Why use "However" here? And why the use of echelle grating has played an essential role in improving the characteristics of the spectrometer?
3. Line 77. "foreign scientific"? Maybe "international" is correct.
4. Line 92. The full name of "UT" has been given in the last paragraph.
5. Line 108. "Professor" should be "professor".
6. "To detect the target infrared radiation, the spectrometer must have a high spectral resolution", why?
7. Line 119. Please explain "n".
8. Line 126. How to reach "real-time detection"?
9. Line 129. I can not find the example of the rocket tail jet in the paper.
10. Line 157. As shown in "Figure 2"?
11. Line 187. Please check the sentence "Figure3. Attenuation coefficients of representative infrared materials."
12. Line 208. Please check the sentence "Optical path difference [57]."
13. Line 209. is shown in Figure 4.
14. Line 234-238. The long sentence is difficult to understand. Please revise it.
15. Line 324. Please add a space before "(c)". Add "," before "(d)". Similar errors appear in Figures 10, 12, and 14.
16. Line 424. 7862 nm.
17. Line 430. Add a space before "We".
18. Line 431-434. The long sentence is difficult to understand. Please revise it.
19. Line 458. How to understand " the remote sensing of tail flames"?
20. Some real experiments or imaging tests can better verify the performance of the developed spectrometer.
Round 2
Reviewer 1 Report
From my point of view, the paper still need some major improvements. I think that in the present form the paper should be rejected.
Page 7. I think that the introduction of the APB light path function in this context is not necessary, since the aim of the paper is not the discussion of the basic principle of toroidal gratings (well known since 1960s), but the use of a toroidal grating for an echelle spectrometer. The authors should simplify the presentation, it would be sufficient to comment on the additional degree of freedom given by the free sagittal radius and report the two focusing equations 10-11. Eq.s 6-9 can be omitted.
Page 7 and Page 9. “Toroidal gratings can correct astigmatism through different curvature radii in two directions, variable-pitch concave gratings can convert most of the aberrations through additional tiny coefficients, and toroidal gratings have both of the above” There are two equal sentences not inserted in the context of the discussion. Correct the sentences and remove the reference to variable-pitch concave gratings, it is not clear to the reader why variable-pitch gratings are cited. The authors present uniform-line-spaced toroidal gratings, therefore the discussion should be limited to those gratings.
Page 9. “Determine the radii of curvature in the sagiittal and meridional directions of the toroidal grating. Calculate the meridional focal length of the toroidal grating used to correct for astigmatism. Finally, the calculated parameters are input into the optical design software Zemax for ray tracing and optimization to verify the performance of the optical path system.” This is not a clear sentence, is just a list of items. This paragraph on the design procedure should be completely reformulated. SInce this really the core of the paper (i.e., the use of a "toroidal grating") the procedure should be very clearly explained. The paragraph needs to mbe expanded.
Page 11. “The grating dispersion direction is different from the vertical-grating-dispersion direction, which can not only realize the separation of orders but also realize the convergence of light beams and simultaneously meet the requirements of high energy utilization and high imaging quality.” The sentence is not clear, please reformulate introducing in a clear way when you refer to the echelle immersion grating and when you refer to the toroidal grating.
Page 13. The two radii of the toroidal grating are very close. The authours should comment on feasibility of such grating. Thermal design for infrared lenses (which is not the main task of the paper) is presented in details, while no commenti s made on the feasibility (and also cost) of the grating. Feasibility: please, comment on the realization of a surface with two radii very close; comment also on the precision that is needed for such a surface. Cost: such a grating would be much more expensive than a spherical one (at least one order of magnitude), comment on the advantages that may compensate the higher cost. Again, this is a very important point for the paper, that is completely omitted in the present version.
Thermal design of the infrared lens. The paper is focused on the use of a toroidal grating (or at least this is highlighted in the title). The thermal design of the infrared lens is not directly linked to the use of a toroidal grating, but it is a more general problem. Are there some particular problems related to the use of the toroidal grating that have been solved in the design? This is not explained in the text.
Conclusions
The paper is focused on the use of a toroidal grating, but the advantages of using a toroidal surface are not clearly resumed. In the final comments, the authors should clearly resume what are the advantages of using a toroidal surface in the final focusing properties.
Reviewer 2 Report
The author has revised most issues I proposed.
Author Response
Revised according to reviewer comments.
Round 3
Reviewer 1 Report
I thank the authors for answering to all the questions/observations.
In the present form, the paper is worth to be published.